# Does delaying discharge from intensive care until after tracheostomy removal affect 30-day mortality? Propensity score matched cohort study

Sarah Vollam [1], David A Harrison,[2] J Duncan Young,[1] Peter J Watkinson [1]

[1]Nuffield Department of Clinical Neurosciences, Oxford University, Oxford, UK
[2]Intensive Care National Audit and Research Centre, London, UK

**Correspondence to**
Mrs Sarah Vollam;
sarah.vollam@ndcn.ox.ac.uk

## ABSTRACT

**Objective** To investigate the short-term mortality effect of discharge from an intensive care unit (ICU) with a tracheostomy in place in comparison to delaying discharge until after tracheostomy removal.

**Design** A propensity score matched cohort study using data from the TracMan study.

**Setting** Seventy-two UK ICUs taking part in the TracMan study, a randomised controlled trial comparing early tracheostomy (within 4 days of critical care admission) with deferred tracheostomy (after 10 days if still indicated).

**Participants** 622 patients who underwent a tracheostomy while in the TracMan study between November 2004 and November 2008. 144 patients left ICU with a tracheostomy. 999 days of observation from 294 patients were included in the control pool.

**Interventions** We matched patients discharged with a tracheostomy in place 1:1 with patients who remained in an ICU until either their tracheostomy was removed or they died with the tracheostomy in place. Propensity models were developed according to discharge destination, accounting for likely confounding factors.

**Primary outcome measure** The primary outcome was 30-day mortality from the matching day. For the 'discharged with a tracheostomy' group, this was death within 30 days after the discharge day. For the 'remained in ICU' group, this was death within 30 days after the matched day.

**Results** 22 (15.3%) patients who left ICU with a tracheostomy died within 30 days compared with 26 (18.1%) who remained in ICU (relative risk 0.98, 95% CI 0.43 to 2.23).

**Conclusion** Keeping patients on an ICU to provide tracheostomy care was not found to affect mortality. Tracheostomy presence may indicate a higher risk of mortality due to underlying diseases and conditions rather than posing a risk in itself.

The TracMan trial was registered on the ISRCTN database (ISRCTN28588190).

## INTRODUCTION

Patients who need prolonged artificial ventilation on an intensive care unit (ICU) commonly have a tracheostomy placed.[1–4] Perceived benefits include decreased sedation use, with increased patient comfort and

### Strengths and limitations of this study

► The data used come from a clinical trial using strategies to ensure a robust dataset for analysis.
► The TracMan study was closed in 2008, limiting generalisability with current practice.
► Even with a database of 909 patients from a 72-centre study of tracheostomy timing, we were only able to generate 144 cases, limiting the power of the study to detect an effect of remaining on an intensive care unit (ICU) for tracheostomy care.
► A total of 622 patients were included in the TracMan trial, allowing a large pool of participants from multiple sites and a close match for almost all patients.
► Despite limitations, the TracMan database is a rich source from which to further our knowledge about patients who undergo a tracheostomy while on an ICU, and we are not aware of a similar prospective database.

mobility.[5] In some hospitals, patients who no longer require a ventilator but still need a tracheostomy stay on an ICU until the tracheostomy is removed. In other hospitals, patients with a tracheostomy leave the ICU and receive care on general wards.

Patients leaving an ICU with a tracheostomy have higher in-hospital mortality than those discharged without a tracheostomy.[1 6 7] A tracheostomy may simply be a marker of a group of patients at greater risk of death from other causes (including the illness or condition that necessitates the tracheostomy). Alternatively, ward tracheostomy care may increase mortality (because, for example, there is insufficient skilled care to avoid direct tracheostomy-related deaths). Discovering which of these explains the high in-hospital mortality rates would change practice. Only discharging patients to the ward after decannulation would reduce mortality if ward tracheostomy care is insufficient. This would need a substantial change in current practice. If the high mortality is caused by



other factors associated with the continued tracheostomy requirement, interventions should target these factors. A randomised controlled study where half of the patients are kept on an ICU until decannulation would resolve this question. However, the costs of such a study would be very high as ICU bed capacity would have to increase to accommodate the intervention. A study using propensity score matching to investigate this approach is therefore a compelling first step.

Seventy-two ICUs in the UK took part in the TracMan randomised controlled trial.[8] The trial compared early tracheostomy (within 4 days of critical care admission) with deferred tracheostomy (after 10 days if still indicated). The study generated a tightly defined, detailed database. This allowed us to undertake a propensity score matched cohort study. We investigated the effect of delaying discharge from an ICU until after tracheostomy removal (or death) in comparison to discharge from ICU with a tracheostomy in place.

## MATERIALS AND METHODS
### Data
We extracted information from the TracMan trial database. Nine hundred and nine adult patients from 72 ICUs in the UK participated. Written informed consent or a signed consultee agreement from the patients' legal representative/welfare guardian was obtained. The study was approved by Scotland A REC, reference 04/MRE00/43. All participants had received artificial ventilation for less than 4 days at study entry. The lead clinician predicted they would need at least seven more days of artificial ventilation. Discharge decision-making and provision of tracheostomy support on the ward were according to local usual practice. Detailed inclusion criteria and other study details are published.[8] There were no missing data.

### Patient and public involvement
Patient and public involvement was not sought for this study using a clinical trial database.

### Definitions
We included patients who received a tracheostomy in either arm of the TracMan study. We excluded patients randomised in error or withdrawn from the trial before 30 days. We also excluded patients transferred to another ICU or discharged to another hospital or home. We defined two groups. The 'discharged with a tracheostomy' group had left an ICU with a tracheostomy in place. They went to a ward or a high-dependency unit (HDU) in the same hospital. The 'remained in ICU' group had either been decannulated before they left the ICU or died in the ICU with the tracheostomy in place.

### Development of propensity models
We chose propensity matching to produce a 'remained in ICU' population as similar as possible to the population discharged with a tracheostomy. This prioritised matching

the populations above the individual matching of patients (as would occur with individual patient matching). We developed two propensity models using logistic regression. One model was for discharge to a general ward with a tracheostomy in place. The other model was for discharge to an HDU with a tracheostomy in place. We chose potential confounders from those available within the TracMan dataset. The original TracMan dataset was chosen by an expert trial team and underwent stringent external review. We undertook a literature review to inform the choices made. The following variables were included in the propensity model (as potential confounders of the relationship between leaving an ICU with a tracheostomy in place and short-term mortality):

► Age on admission to the ICU
► Acute Physiology And Chronic Health Evaluation (APACHE) II Acute Physiology Score (APS) on ICU admission
► Surgical status (admission from an operating theatre following elective/scheduled surgery, admission from theatre following emergency/urgent surgery or non-surgical)
► Underlying reason for mechanical ventilation (neurological/non-neurological)
► Number of days since tracheostomy (up to and including the day of discharge for cases or matched day for controls)
► Number of days receiving an intravenous bolus dose or an infusion of drugs primarily for sedation (up to and including the day of discharge for cases or matched day for matched controls)
► Receiving an intravenous bolus dose or an infusion of drugs primarily for sedation on the day of discharge for cases or matched day for matched controls
► Number of days of support (up to and including the day of discharge for cases or matched day for matched controls) of the following organ systems:
  – Advanced respiratory support (invasive artificial ventilation; extracorporeal gas exchange)
  – Basic respiratory support (mask continuous positive airway pressure, non-invasive ventilation, >50% inspired oxygen, recent extubation following prolonged mechanical ventilation, intubation to protect the airway, physiotherapy to clear secretions at least 2 hourly)
  – Circulatory support (vasoactive drug infusion, circulatory instability due to hypovolaemia, patients resuscitated following cardiac arrest and intra-aortic balloon pump in place)
  – Neurological support (central nervous system depression sufficient to prejudice the airway and invasive neurological monitoring)
  – Renal support (acute renal replacement therapy)
► Organ support (as previously mentioned) received on the day of discharge for cases or matched day for controls.

## Matching

We matched patients discharged with a tracheostomy (cases) 1:1 with the patient (and day) from the remained in ICU group (controls). We matched to the patient with the closest propensity score (predicted log odds from the propensity model). We matched patients up to a difference of 1 SD of the propensity score. We used the ward or HDU propensity model according to the patient's discharge destination.

To ensure comparability between the two groups, the 'remained in ICU' group were required to have had the tracheostomy for at least two days by the matched day; have not received advanced respiratory support on the day following the matched day; and (for controls matched to patients discharged to a ward with the tracheostomy) have not received neurological support or renal support on the day following the matched day. We give our definitions of organ support above.

## Outcome

The primary outcome was 30-day mortality from the matching day. For the 'discharged with a tracheostomy' group, this was death within 30 days after the discharge day. For the 'remained in ICU' group, this was death within 30 days after the matched day.

## Assessment of balance

We compared cases and controls before and after matching for their age, sex, APACHE II APS, overall APACHE II Score, surgical status, underlying reason for mechanical ventilation, number of days since tracheostomy, number of days receiving sedatives, number of days of organ support and organ support received on matched day. We compared continuous variables with mean and SD. We compared categorical variables with number and per cent. We compared balance with the standardised difference. We defined the standardised difference as the absolute value of the difference in means divided by the pooled SD.[9] In addition, we used quantile–quantile (QQ) plots to assess the balance across the full distribution for age, APACHE II Score and propensity score. We assessed balance for the full cohort and split by the case's destination following discharge (ward or HDU).

## Statistical analysis

We estimated the relative risk of 30-day mortality for cases compared with matched controls by Poisson regression, conditional on the matched data.[10] We estimated standard errors with the non-parametric bootstrap.[11] We calculated relative risks both unadjusted and adjusted for age, APACHE II APS and respiratory support on the matched day, to control for any remaining imbalance in these risk factors. We evaluated any difference in effect between patients discharged to a ward and those discharged to an HDU by introducing an interaction term in the Poisson regression model. We performed all analyses using STATA/SE V.13.0.

## RESULTS

A total of 909 patients were enrolled in the TracMan trial between November 2004 and November 2008. We excluded 10 patients randomised in error or withdrawn from the trial before 30 days. Of the remaining 899 patients, 622 (69.2%) underwent a tracheostomy. A total of 174 deaths in ICU occurred in this group. One hundred forty-four patients had a tracheostomy in place when they left the ICU. Seventy patients (11.3%) were discharged to a ward in the same hospital. Seventy-four (11.9%) were discharged to an HDU in the same hospital. We included 999 days of observation from 294 patients in the control pool. For the ward control pool, we included 960 days of observation from 292 patients (as we did not include days where patients received neurological or renal support). Figure 1 shows the selection process for cases and controls.

Table 1 shows balance statistics for the full cohort, before and after matching. Balance statistics for those discharged to a ward and those discharged to an HDU are included in the online supplementary material table S1 and S2. There was imbalance in many covariates before matching (as indicated by standardised difference values above 10). Balance improved after matching. Figure 2 shows QQ plots for age, APACHE II Score and propensity score. The QQ plot of propensity score balance shows almost all patients were closely matched.

Remaining in an ICU for tracheostomy care did not statistically affect mortality in comparison to care outside an ICU (table 2).

## DISCUSSION

There is little doubt that hospital mortality after leaving an ICU is high in patients with a new tracheostomy.[1 2 7] Our study supports these previous findings. Patients leaving the ICU with a tracheostomy had a 30-day mortality nearly twice the 9.5% in-hospital mortality for all patients discharged from UK ICUs at the time.[12] There are two possible explanations for this high mortality. There may be deficiencies in tracheostomy care outside ICU, which staying in an ICU would avoid. Alternatively, having a tracheostomy when leaving an ICU may simply mark a higher-risk group. To separate these two possibilities, we propensity score matched patients who left an ICU with a tracheostomy with patients who remained in an ICU with a tracheostomy. The TracMan database contained high-quality daily patient information. These data allowed us to match on time and treatment periods, as well as static variables. Patients required similar levels of support when matched. We found no significant difference in post-ICU 30-day mortality. This remained true when we separated patients discharged to a general ward and those discharged to an HDU.

Our propensity score matched cohort design limits the study findings. Although balance was improved between groups after matching, imbalances remained. This was always likely, given the number of factors matched. Our

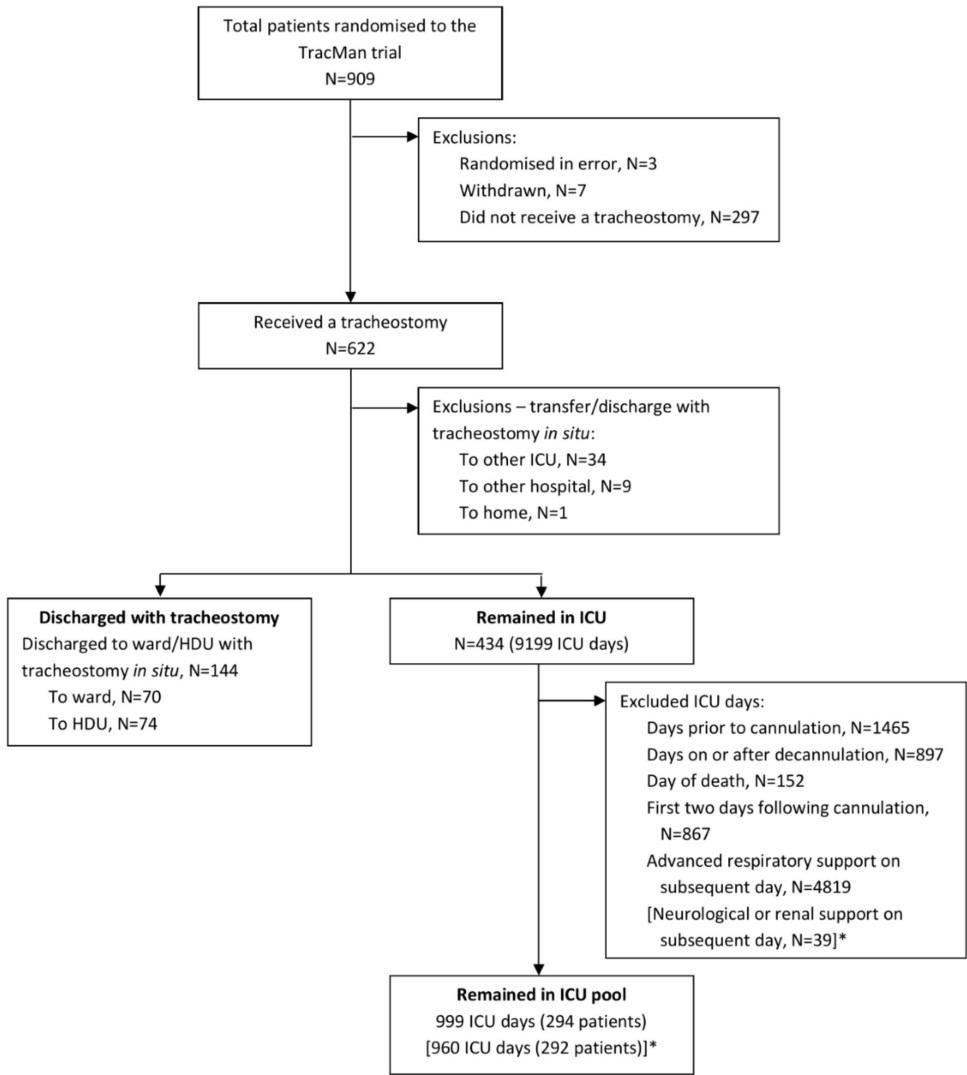

**Figure 1** Case and control selection flow diagram. *For comparison with patients discharged to the ward. HDU, high-dependency unit; ICU, intensive care unit.

results may be confounded by an unknown and unrecorded variable that alters both the likelihood of leaving an ICU with a tracheostomy and subsequent death. This is a risk in all similarly designed studies. Our detailed database, allowing us to match on multiple variables, should have minimised this risk. The original TracMan study did not collect detailed information on post-tracheostomy care. However, the large number of sites taking part in the study would suggest that this care represented common UK practice.

The data used come from a clinical trial, in which no deaths were attributed to the tracheostomy procedure. Variables were collected using a comprehensive set of definitions to ensure consistency. Out-of-range or missing data were checked during the trial with source documents. These strategies provided a robust dataset for analysis. However, the database does not include all patients who had a tracheostomy at the centres during the study period. Some patients will have undergone a tracheostomy outside the study. Patients in a clinical trial may experience closer monitoring, and therefore

different care, than in usual practice.[13] However, the TracMan study did not require ward or high-dependency unit follow-up by research staff. The risk that researcher staff directly affected ward care appears small.

Even with a 909-patient database available from a 72-centre study of tracheostomy timing, we were only able to generate 144 cases, limiting the power of the study to detect an effect of remaining on an ICU for tracheostomy care. The wide CIs around our estimates mean we cannot exclude a clinically relevant effect. Cases were matched 1:1 with controls. Matching at one case to two controls may have increased power slightly. Conversely, as we used replacement, this would also have increased the duplication of patients in the control group.

Although these concerns place some limitations on the generalisability of our study, the results remain important. Despite being over 10 years old, the TracMan database is a rich source from which to further our knowledge about patients who undergo a tracheostomy while on an ICU. We are not aware of a similar prospective database. Our study benefits from the large number (622) of patients

**Table 1** Balance in potential confounders before and after matching: full cohort

| | Before matching | | | After matching | | |
|---|---|---|---|---|---|---|
| | Discharged with tracheostomy (n=144) | Remained in ICU (n=999*) | Standardised difference | Discharged with tracheostomy (n=144) | Remained in ICU (n=144†) | Standardised difference |
| Age (years), mean (SD) | 64.5 (12.5) | 62.7 (13.1) | 14.1 | 64.5 (12.5) | 65.4 (11.2) | 7.2 |
| Male, n (%) | 82 (56.9) | 644 (64.5) | 15.4 | 82 (56.9) | 104 (72.2) | 32.2 |
| APACHE II APS, mean (SD) | 14.4 (5.8) | 13.4 (6.5) | 16.2 | 14.4 (5.8) | 13.7 (6.7) | 11.4 |
| APACHE II Score, mean (SD) | 19.2 (6.8) | 17.9 (7.0) | 19.3 | 19.2 (6.8) | 18.8 (7.3) | 5.6 |
| Surgical status: elective/scheduled, n (%) | 8 (5.6) | 51 (5.1) | 2.0 | 8 (5.6) | 10 (6.9) | 5.7 |
| Surgical status: emergency/urgent, n (%) | 24 (16.7) | 140 (14.0) | 7.4 | 24 (16.7) | 25 (17.4) | 1.8 |
| Neurological reason for ventilation, n (%) | 18 (12.5) | 100 (10.0) | 7.9 | 18 (12.5) | 19 (13.2) | 2.1 |
| Days since tracheostomy, mean (SD) | 14.6 (10.4) | 17.2 (13.6) | 21.4 | 14.6 (10.4) | 14.1 (10.9) | 4.6 |
| Days receiving sedatives, mean (SD) | 8.1 (6.9) | 9.1 (6.9) | 14.6 | 8.1 (6.9) | 7.8 (5.7) | 4.6 |
| Days of advanced respiratory support, mean (SD) | 15.3 (10.9) | 19.0 (14.1) | 29.2 | 15.3 (10.9) | 15.1 (11.9) | 1.8 |
| Days of basic respiratory support, mean (SD) | 5.9 (9.5) | 5.5 (9.5) | 4.5 | 5.9 (9.5) | 6.8 (11.9) | 8.3 |
| Days of circulatory support, mean (SD) | 3.4 (3.7) | 4.3 (4.7) | 21.2 | 3.4 (3.7) | 3.2 (3.2) | 5.0 |
| Days of neurological support, mean (SD) | 3.5 (6.6) | 2.4 (5.3) | 18.9 | 3.5 (6.6) | 3.0 (5.2) | 10.0 |
| Days of renal support, mean (SD) | 1.2 (4.2) | 2.3 (6.6) | 19.9 | 1.2 (4.2) | 0.9 (3.0) | 8.8 |
| Receiving sedatives on final day, n (%) | 8 (5.6) | 92 (9.2) | 14.0 | 8 (5.6) | 7 (4.9) | 3.1 |
| Advanced respiratory support on final day, n (%) | 29 (20.1) | 322 (32.2) | 27.7 | 29 (20.1) | 21 (14.6) | 14.7 |
| Basic respiratory support on final day, n (%) | 89 (61.8) | 612 (61.3) | 1.1 | 89 (61.8) | 90 (62.5) | 1.4 |
| Circulatory support on final day, n (%) | 0 (0) | 32 (3.2) | 25.7 | 0 (0) | 0 (0) | N/A |
| Neurological support on final day, n (%) | 6 (4.2) | 22 (2.2) | 11.2 | 6 (4.2) | 2 (1.4) | 16.9 |
| Renal support on final day, n (%) | 0 (0) | 37 (3.7) | 27.7 | 0 (0) | 0 (0) | N/A |

*999 days of observation from 294 patients.
†84 unique patients after matching with replacement.
APACHE, Acute Physiology And Chronic Health Evaluation; APS, Acute Physiology Score; ICU, intensive care unit; N/A, not applicable.

who underwent a tracheostomy within the TracMan trial and the prospective daily information gathered. These factors allowed a large pool of patients from which to match. As a result, almost all patients were closely matched. The scale of the database allowed us to include substantially more patients from many more centres than has previously been possible.

A previous multicentre study took a different propensity-based approach. They compared 141 patients leaving an ICU with a tracheostomy in place with 60 leaving after tracheostomy removal.[2] No difference in hospital mortality *after leaving* an ICU was found. Our study assesses the effect of tracheostomy care *in an ICU* in comparison with care outside an ICU at an equivalent point in patients' recovery. Our approach has the advantage that mortality while on ICU undergoing tracheostomy care is taken into account. Neither approach suggests that mortality is increased by tracheostomy care outside an ICU. The two propensity-based studies have findings that contrast with previous research.[6 7] The findings demonstrate the

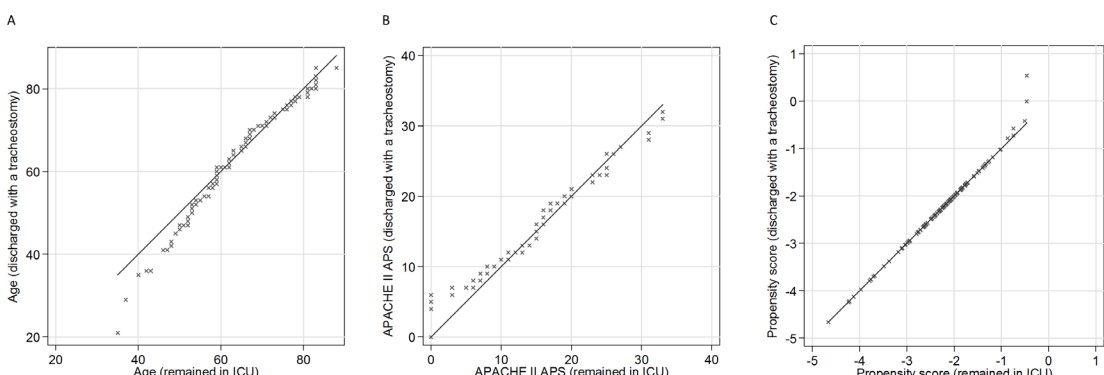

**Figure 2** Quantile–quantile plots of balance in (A) age, (B) Acute Physiology And Chronic Health Evaluation II Acute Physiology Score and (C) propensity score. ICU, intensive care unit.

**Table 2** Outcome of 30-day mortality following discharge with a tracheostomy in place compared with those who remained in an ICU

| | 30-day mortality, deaths (%) (95% CI) | | | | | |
|---|---|---|---|---|---|---|
| | Discharged with tracheostomy | Remained in ICU | Unadjusted RR (95% CI) | P value | Adjusted RR* (95% CI) | P value |
| Overall (n=144) | 22 (15.3) (10.3 to 22.0) | 26 (18.1) (12.6 to 25.1) | 0.85 (0.49 to 1.47) | 0.55 | 0.98 (0.43 to 2.23) | 0.96 |
| Discharged to a ward (n=70) | 13 (18.6) (11.2 to 29.2) | 11 (15.7) (9.0 to 26.0) | 1.18 (0.48 to 2.89) | 0.48† | 1.46 (0.45 to 4.78) | 0.37† |
| Discharged to an HDU (n=74) | 9 (12.2) (6.5 to 21.5) | 15 (20.3) (12.7 to 30.8) | 0.60 (0.11 to 3.30) | | 0.61 (0.15 to 2.57) | |

*Adjusted for age and Acute Physiology And Chronic Health Evaluation II Acute Physiology Score and respiratory support on final day.
†P value for test of interaction with destination following discharge.
HDU, high-dependency unit; ICU, intensive care unit; RR, relative risk.

importance of taking imbalances in other risk factors into account.

Two other information sources support the view that the mortality attributable to the tracheostomy and its care after leaving an ICU may be low. A study reviewing 453 incidents related to tracheostomies in patients treated on hospital wards reported to the UK National Patient Safety Agency over two years[14] found only 15 (3%) where the incident was associated with the death of a patient. In a national audit, only 5 of 518 (1%) ward patients with a tracheostomy were classified as having suffered harm from tracheostomy-related problems. Only one of these suffered a cardiac arrest.[15]

The question of where to provide care for patients who have had a tracheostomy placed on an ICU might best be answered by a randomised controlled trial. However, the incidence of events in our study and those of others suggest a very large (and so expensive) trial would be required.[2] At present, a propensity-matched approach may provide the highest level of evidence practically available.

Recommendations for improving ward outcomes for patients leaving an ICU with a tracheostomy have focused on tracheostomy care.[15–17] Ensuring appropriate tracheostomy care and training for tracheostomy-related emergencies is clearly important. In undertaking a large multicentre randomised controlled trial, a limited number of outcomes can be collected. Individual hospital tracheostomy care was not collected in the study. Other problems underlie the continued need for a tracheostomy on the ward. Our findings indicate that remaining in an area of high staff ratio and skill mix, suggesting safe tracheostomy management, did not improve mortality. Therefore, mortality may be more related to the underlying diseases and conditions than the tracheostomy itself. We need better understanding of these issues to improve outcomes for these vulnerable patients.

## CONCLUSION

In a propensity-based analysis of a large prospective dataset, 30-day mortality was high in patients who could be discharged from an ICU with a tracheostomy in place. However, keeping patients on an ICU to provide tracheostomy care did not affect mortality in comparison to ward-based care. Tracheostomy presence may indicate a higher risk of mortality due to underlying diseases and conditions rather than posing a risk in itself. This should be considered when planning care for patients with a tracheostomy.

**Acknowledgements** The full list of TracMan trial collaborators has been published and their major contribution to this study is acknowledged. The TracMan trial was registered on the ISRCTN database (ISRCTN28588190).

**Contributors** DY, PJW and SV conceived the original idea. SV and DY prepared the manuscript with revisions from PJW and DAH. All authors read and approved the final manuscript. All authors had full access to all of the data (including statistical reports and tables) in the study and take responsibility for the integrity of the data and the accuracy of the data analysis.

**Funding** The work described in this paper was funded by the National Institute for Health Research (NIHR) Oxford Biomedical Research Centre and by the Intensive Care National Audit and Research Centre. The views expressed are those of the authors and not necessarily those of the NHS, the NIHR or the Department of Health. The original TracMan study was funded by the UK Intensive Care Society and the Medical Research Council.

**Competing interests** PJW codeveloped the System for Electronic Notification and Documentation, for which Sensyne Health has purchased a sole licence. The company has a research agreement with the University of Oxford and royalty agreements with Oxford University Hospitals NHS Trust and the University of Oxford. Sensyne Health may in the future pay PJW personal fees. PJW was chief medical officer for Sensyne Health.

**Patient and public involvement** Patients and/or the public were not involved in the design, conduct, reporting or dissemination plans of this research.

**Patient consent for publication** Not required.

**Provenance and peer review** Not commissioned; externally peer reviewed.

**Data availability statement** Deidentified data may be made available on reasonable request to the corresponding author.

**ORCID iDs**
Sarah Vollam http://orcid.org/0000-0003-2835-6271
Peter J Watkinson http://orcid.org/0000-0003-1023-3927

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
