## [Reviewer comments · BMJ Open]

ARTICLE DETAILS

TITLE (PROVISIONAL)	Does delaying discharge from intensive care until after tracheostomy removal affect 30-day mortality? A propensity score matched cohort study.
AUTHORS	Vollam, Sarah; Harrison, David A; Young, Duncan; Watkinson, Peter J

VERSION 1 – REVIEW

REVIEWER	Professor Patrick J Bradley Queens Medical Centre Nottingham United Kingdom
REVIEW RETURNED	25-Feb-2020

GENERAL COMMENTS	An admirable project and the use of material from 72 UK ICU would seem appropriate to suggest the conclusion reached - however the volume of practice (size, numbers and type of patient trough-put) may have biased those patients who were Discharge with Tracheostomy vs. Remain with Tracheostomy("DwT" vs. "RwT" thus HDU/Ward care may have been more conducive to early discharge from ITU? However there has been no discussion on the several factors: What was the cause of death specifically related to the placement of the tracheostomy? What was the expertise of the staff that cared for the group "DwT"? What was the designation of the ward care -- and the proportions discharged to ward vs. HDU? How was selection of patients who were DwT" vs. "RwT"? How long had patients been "off ventilator support" before such a decision had been made for final placement? Of the 72 ICU's partaking of TracMan project was there a dominance of any unit or units that biased the group DwT (RwT 294 vs. DwT 144)? Comparing the groups how many patients subsequently were decannulated (removal of tracheostomy) within 5 - 7 days in both groups? The questioned posed is important but requires a major change of current practice within the NHS -- it would seem appropriate that "all patients with a tracheostomy not respiratory support" would be best nursed by a specialist team trained in tracheostomy care in a specialist environment, and not even requiring HDU! Avoidable deaths specifically related to the tracheostomy site or the tube itself while "rare" is seldom reported as the cause of death?
--

REVIEWER	Piero Ceriana ICS Maugeri - Pavia -Italy
REVIEW RETURNED	26-Feb-2020

GENERAL COMMENTS	The paper addresses an issue that has been debated and is not fully resolved. Therefore the contribution given by Volla and coworkers must be considered with interest. However, reading the paper, there is a point not fully clear to me: if I am not wrong, some patients with tracheostomy were discharged to ward or high dependency unit, while other patients remained in the ICU. Since this is a retrospective analysis and the allocation of patients was not based on a randomization order, I ask to authors to clarify the criteria or the guidelines of the Units about when and how to discharge patients, otherwise it could be argued that less severe patients were discharged and the unstable ones remained. Are the criteria based on the availability of beds outside the ICU? were some patients discharged urgently in order to make an ICU bed available for an acute and urgent case? all the items should be discussed and made clear.
---

REVIEWER	Federica Porcaro Bambino Gesù Children's Hospital, Italy
REVIEW RETURNED	26-Feb-2020

GENERAL COMMENTS	The Authors studied the mortality risk in two groups of patients (tracheostomized patients discharged from an ICU and patients remained in an ICU until tracheostomy removal or death with the tracheostomy in place). They concluded that the 30-day mortality was high in patients discharged from an ICU with a tracheostomy in place. In addition to the fragility of tracheostomized patients, do you think that inadequate tracheostomy management has affected mortality? Is there a training in the management of tracheostomy for patients' family members or caregivers in your hospital?
---

VERSION 1 – AUTHOR RESPONSE

Reviewer: 1

- An admirable project and the use of material from 72 UK ICU would seem appropriate to suggest the conclusion reached - however the volume of practice (size, numbers and type of patient through-put) may have biased those patients who were Discharge with Tracheostomy vs. Remain with Tracheostomy ("DwT" vs. "RwT" thus HDU/Ward care may have been more conducive to early discharge from ICU?

Thank you for your positive comments on this study.

As we state under “development of propensity models”, we chose propensity matching to produce a “remained in ICU” population as similar as possible to the population “discharged with a tracheostomy”. We matched on a large number of clinically relevant variables. Although it is possible that residual bias may have remained (as we acknowledge in the Discussion section, paragraph 2), we state in the Results section, paragraph 2 that “the QQ plot of propensity score balance (figure 2) shows almost all patients were closely matched”.

- However there has been no discussion on the several factors: What was the cause of death specifically related to the placement of the tracheostomy?

We agree this is an important detail. The original study found no deaths were directly procedure

related – we have included this in the Discussion section (paragraph three).

- What was the expertise of the staff that cared for the group "DwT"?

The TracMan study took place in 72 ICUs in the UK. We therefore believe the expertise of staff would be representative of common practice. We have commented on this in the Discussion section (paragraph 2) and made clear that tracheostomy care followed local practice in Materials and Methods (Data sub-section, paragraph 1).

- How was selection of patients who were DwT" vs. "RwT"? How long had patients been "off ventilator support" before such a decision had been made for final placement?

We have clarified in the first paragraph of Materials and Methods (Data sub-section) that discharge decisions followed local practice. We explain in Materials and Methods (sub-section Matching, paragraph 1) that “to ensure comparability between the two groups, the “remained in ICU” group were required to: have had the tracheostomy for at least two days by the matched day; have not received advanced respiratory support on the day following the matched day”

- Of the 72 ICU's partaking of TracMan project was there a dominance of any unit or units that biased the group DwT (RWT 294 vs. DwT 144)?

Operational differences between practitioners or units may explain why patients were in the RWT or DwT group. It is such differences, given propensity-matched patients, that we are exploiting to examine the question posed.

- Comparing the groups how many patients subsequently were decannulated (removal of tracheostomy) within 5 - 7 days in both groups?

We acknowledge that this would have been interesting, however, in the nature of a large pragmatic randomised controlled trial only a limited number of outcome measures can be collected, of which this was not one.

- The questioned posed is important but requires a major change of current practice within the NHS -- it would seem appropriate that "all patients with a tracheostomy not respiratory support" would be best nursed by a specialist team trained in tracheostomy care in a specialist environment, and not even requiring HDU! Avoidable deaths specifically related to the tracheostomy site or the tube itself while "rare" is seldom reported as the cause of death?

Thank you for recognising the importance of this question. Commenting on cause of death documentation is outside of the scope of this paper, but avoidability of post-ICU death is a key focus of our ongoing research.

Reviewer: 2

- The paper addresses an issue that has been debated and is not fully resolved. Therefore the contribution given by Vollam and coworkers must be considered with interest. However, reading the paper, there is a point not fully clear to me: if I am not wrong, some patients with tracheostomy were discharged to ward or high dependency unit, while other patients remained in the ICU. Since this is a retrospective analysis and the allocation of patients was not based on a randomization order, I ask to authors to clarify the criteria or the guidelines of the Units about when and how to discharge patients, otherwise it could be argued that less severe patients were discharged and the unstable ones remained. Are the criteria based on the availability of beds outside the ICU ? were some patients

discharged urgently in order to make an ICU bed available for an acute and urgent case ? all the items should be discussed and make clear.

Thank you for your interest in our work. Discharge decisions were made following local practice, in keeping with a large pragmatic multicentre randomised controlled trial. To investigate this question, we conducted an additional analysis separating HDU and ward discharge destinations. This is included in the supplementary information for reference.

Reviewer: 3

- The Authors studied the mortality risk in two groups of patients (tracheostomized patients discharged from an ICU and patients remained in an ICU until tracheostomy removal or death with the tracheostomy in place). They concluded that the 30-day mortality was high in patients discharged from an ICU with a tracheostomy in place.

In addition to the fragility of tracheostomized patients, do you think that inadequate tracheostomy management has affected mortality?

We discuss this issue in the Discussion section, paragraphs 6- 9. In summary, we believe that the combination of our work and that of others suggests that mortality attributable to the tracheostomy and its care after leaving an ICU may be low. Our work adds that remaining in ICU until the tracheostomy is removed is not associated with a decreased mortality. We have emphasised this point in paragraph 9.

- Is there a training in the management of tracheostomy for patients' family members or caregivers in your hospital?

Although our hospital both provides care and training for tracheostomy management from our ICU, the study was conducted in 72 ICUs and so practice will have varied across these. We believe this is a considerable strength of our study, as it reflects common practice rather than that of a small number of institutions.

VERSION 2 – REVIEW

REVIEWER	Professor Patrick J Bradley Emeritus Professor Otolaryngology, Head and Neck Surgery Nottingham University Hospitals Queens Medical Centre Campus Derby Road Nottingham United Kingdom
REVIEW RETURNED	22-Apr-2020

GENERAL COMMENTS	My questions were answered arising from the first review and required little change to the original manuscript by the authors to enhance the original manuscript.
---

REVIEWER	Piero Ceriana ICS Maugeri IRCCS - Pavia - Italy
REVIEW RETURNED	21-Apr-2020

GENERAL COMMENTS	I have no other comments to add
---------------------------------